# On the Evaluation Framework of Comprehensive Trust for Data Interaction in Intermodal Transport

**Xin Geng [1,2], Yinghong Wen [1,*], Zhisong Mo [3], Peng Dong [2], Fanpeng Kong [2] and Ke Xiong [4]**

[1] Institute of Electromagnetic Compatibility, Beijing Jiaotong University, Beijing 100044, China; 12111054@bjtu.edu.cn
[2] China Railway Information Technology Group Co., Ltd., Beijing 100844, China; dongpeng@sinorail.com (P.D.); kongfanpeng@sinorail.com (F.K.)
[3] China State Railway Group Co., Ltd., Beijing 100038, China; mozhisong@hotmail.com
[4] School of Computer and Information Technology, Beijing Jiaotong University, Beijing 100044, China; kxiong@bjtu.edu.cn
[*] Correspondence: yhwen@bjtu.edu.cn

**Abstract:** Due to the necessity to realize "building a strong transportation nation", the construction of intermodal transportation is based on the information resource integration of diverse transport systems. To ensure the data security during the interaction between different transport modes, as well as the effect of data application, the status of entities and data flow in the network should be supervised throughout. Therefore, an evaluation framework of comprehensive trust is proposed in this paper. With feature analysis of transportation big data, a quality assessment is conducted by three-dimensional metric sets, which is considered as a significant factor of trust measurement. Furthermore, a hierarchical trust structure is put forward to assess the trust of entities in different levels, in terms of the static and dynamic evidence. Furthermore, the visualization of a dynamic global information security state is discussed, based on temporal knowledge graphs. As shown in practical application and simulation analysis, this framework can meet the requirements of data security supervision and lay the foundation of further intelligent management. This research is of great significance to improve the data security level in intermodal transport, and to promote the utilization and sharing of public information resources.

**Keywords:** intermodal transport; data quality; trust assessment; temporal knowledge graph; data security supervision

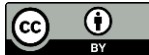

## 1. Introduction

In accordance with national transportation strategies such as 'Belt and Road', and 'building a strong transportation nation', the demand for building a modern comprehensive transportation system has experienced explosive growth in China. As one of the important components, intermodal passenger transport serves as a cornerstone for improving the quality and standards of integrated transportation services, bearing significant importance in expediting structural reforms in transportation service supply and driving the development of modern comprehensive transportation systems. Currently, the implementation of intermodal transportation faces numerous challenges. From the macro point of view of the government, there are socio-economic issues such as accessibility and inclusiveness, impact on the ecological environment, urban development and land planning, and legal and policy safeguards, as well as economic benefits and costs. From the narrow point of view of the passenger, there is a demand for a more convenient transfer service, a more convenient security check service, a simpler ticketing service and a more accurate information service. The resolution of these issues is inseparable from breaking down data silos to achieve an interconnectivity of information resources between different

modes of transportation, such as railways, civil aviation, expressways and urban public transportation. Nowadays, the comprehensive utilization of transportation big data has garnered increasingly widespread attention. In particular, it is necessary to promote integrated data utilization between forms of transport, to realize the vision of route planning with a 'unified map', operating control with a 'unified table', and travelling with a 'unified ticket', so as to better serving the economic and social development of metropolitan areas and improving people's lives [1].

As a strategic asset and business-driving factor, the data resources of transportation enterprises are currently used separately, with the separation of ownership and use right. Consequently, data silos of transportation resources have led to a dilemma in data sharing and exchange that the requirements for secure sharing and development cannot satisfy. Due to the diversity of data sources and the risk of data transmission, cross-entity, cross-domain, and even cross-mode data exchange may suffer from severe issues, such as data leakage, data compliance issues, data loss and security vulnerabilities. Therefore, it is imperative to supervise and conduct real-time monitoring of the security of relevant entities from the overall perspective to ensure the smooth operation of critical information infrastructure concerning intermodal transportation.

There has been some research on the security evaluation of datasets and access entities for cross-domain interaction. In order to prevent data quality from affecting the secondary application, a multi-level assessment framework based on a business scenario is established to objectively evaluate the data quality of intelligent transportation [2]. A service trust assessment model towards a cross-domain alliance of services is proposed, which combines the characteristics of the service collaboration, the characteristics of service entities, various links in service life cycle and inter-domain security policy [3]. Considering the influence brought by the confidentiality, as well as the synchronization of the trusted measurement frequency with the progress, a multilevel dynamic trusted measurement model based on information flow theory is raised [4]. In order to solve the identity credibility problem in cross-domain authentication under the blockchain-based heterogeneous identity alliance infrastructure, the identity credibility evaluation method of alliance member consensus has been designed [5]. With a trust-based logical framework and trust measurement model, the vicious competition of agents in smart cities can be solved, and agents can adopt a cooperative strategy [6].

The various research mentioned above has realized a flexible and scalable trust measurement of cross-domain information resources, but there are still some shortcomings for the scenario of intermodal transportation. Firstly, as for the data quality assessment, the framework should combine with the properties of traffic big data, where the spatiotemporal characteristics should be adequately addressed. Secondly, considering the information collaboration requirements, data quality should be taken as a great factor of data trust. In fact, the outcome of a business analysis task is directly influenced by the quality of input data, especially in complex analysis tasks; thus, it directly affects the security of data applications. For example, decision-making and predictions based on patterns and trends with poor quality or failing to capture real insights and images of current operational status, can lead to unnecessary risks and losses for transportation organizations, and may have effect on the safety and convenience of passengers' travelling experience [7]. Thirdly, the trust measurement should be dynamically updated with data flow between entities in the network environment, considering both static and dynamic evidence for different types of entities. In this way, the data security situation of organizations at all levels can be understood in due course, and the integration of comprehensive supervision of data security realized. Finally, in order to ensure the data security of the whole network, it is necessary to provide an effective visual method to find security threats and judge the trend of data security from a global perspective.

Therefore, in this paper, we propose a comprehensive trust evaluation framework for data interaction entities in intermodal transportation, from the perspective of a data

security supervisor. It takes trust as a metric to conduct a comprehensive security monitoring of integrated data, in order to effectively guard the normal operation and integrity of critical information infrastructure. Building upon this foundation, it enables an understanding of the data security posture at different levels, facilitates the exchange of data flow information in the network environments, assesses data security compliance from a regulatory standpoint, and evaluates data security risk trends from a holistic perspective. Furthermore, the framework is designed to be implemented on the data integrated service platform and the performance of data transmission will not be affected.

The structure of the paper is as follows: Section 2 analyzes the characteristics of data to share between different transportation modes and proposes the hierarchical evaluation framework based on global security needs. Section 3 raises a model based on three dimensions of time, space and content, to assess the quality of transportation data on different aspects. Section 4 introduces a three-layer trust evaluation structure, considering both inherent attributes and historical interaction behaviors. Moreover, to dynamically display the entity performance in the process of cross-mode data interaction, a visualization based on a temporal knowledge graph is adopted in Section 5. Section 6 demonstrates the feasibility and practicality of the framework through practical examples of a passenger services scenario. Finally, in the last section, a conclusion summarizes the entire paper.

## 2. Problem Formulation

### 2.1. Demand of Data Interaction in Intermodal Transportation

Transportation big data generally refers to a dataset generated directly by the management of urban transportation operations, including various types of data related to road traffic and public transport. The broad definition also includes data from industries or fields related to transportation, such as meteorological environment, population planning, mobile communication signaling, etc.; in addition, it counts in traffic condition data provided by mass media, such as text, images, audio, and video provided through Weibo, forums and radio broadcasts, etc. By definition, transportation big data possesses the following characteristics:

A. Huge Resource Volume. Urban transportation generates a massive amount of data continuously, including external data such as weather and environmental monitoring. In large cities like Shanghai, for instance, the structured transportation data alone can exceed 30 GB per day [5], in addition to vast amounts of unstructured data such as road surveillance videos and toll booth photos.

B. Diverse Information Formats. From the perspective of data sources, this includes the data generated directly by data acquisition equipment on-board and off-board, as well as other information systems. Furthermore, the diversity in professional types, wide business scope, and extended service chains contribute to the variety of data types. Taking the railway system as an example, railway big data spans the entire lifecycle from "survey and design—engineering construction—joint debugging and testing—operation and maintenance", covering the entire business chain of vehicles, machinery, engineering, electricity, and vehicles [3,8].

C. Significant Spatiotemporal Characteristics. Traffic network data usually has a temporal and spatial dimension of massive scale; thus, it belongs to typical spatiotemporal big data. For example, railway information resources come from all locomotives, vehicles, and various sensors on infrastructure in more than 600 station segments of 18 railway companies nationwide, exhibiting clear geographical distribution [4]. In addition, with the development of urban transportation, decision-making in traffic management emphasizes the analysis of recent data. It indicates that historical data has a much lower reference value for traffic management and urban planning decision-making compared to recent data. In particular, in applications such as traffic diversion, accident warnings, and route planning, timely and accurate data collection and processing are indispensable.

**D. High Security Requirements.** On one hand, transportation data holds rich value and can provide strong support for emergency traffic planning, enabling rapid response and emergency command in critical situations, contributing to social stability and reducing economic losses [9]. On the other hand, the data of the transport industry involves daily management, operational maintenance, and dispatch arrangements for enterprises. Data leakage can pose severe risks to business secrets and public safety. Therefore, the requirement for resource security remains a high level.

To achieve hub transfers, capacity allocation, and dispatch command in metropolitan intermodal transportation, there is need for systematic research on collaborative transportation organization based on integrated information resources. Meanwhile, passengers have higher expectations for intelligent, comprehensive, flexible, interactive, transparent, and efficient information services, including more accurate information services, convenient access anytime and anywhere, and intelligent queries based on natural language interaction. Thus, the comprehensive utilization of transport network facilities and equipment capacities will differ from the previous single-mode form. Furthermore, research is still needed for the coordination and command of emergency situations in the safety assurance system. The realization of these visions depends on the information collaboration and exchange between different transportation systems. According to a previous research and literature review [6,10], the essential information requested in passenger intermodal transportation is shown in Figure 1.

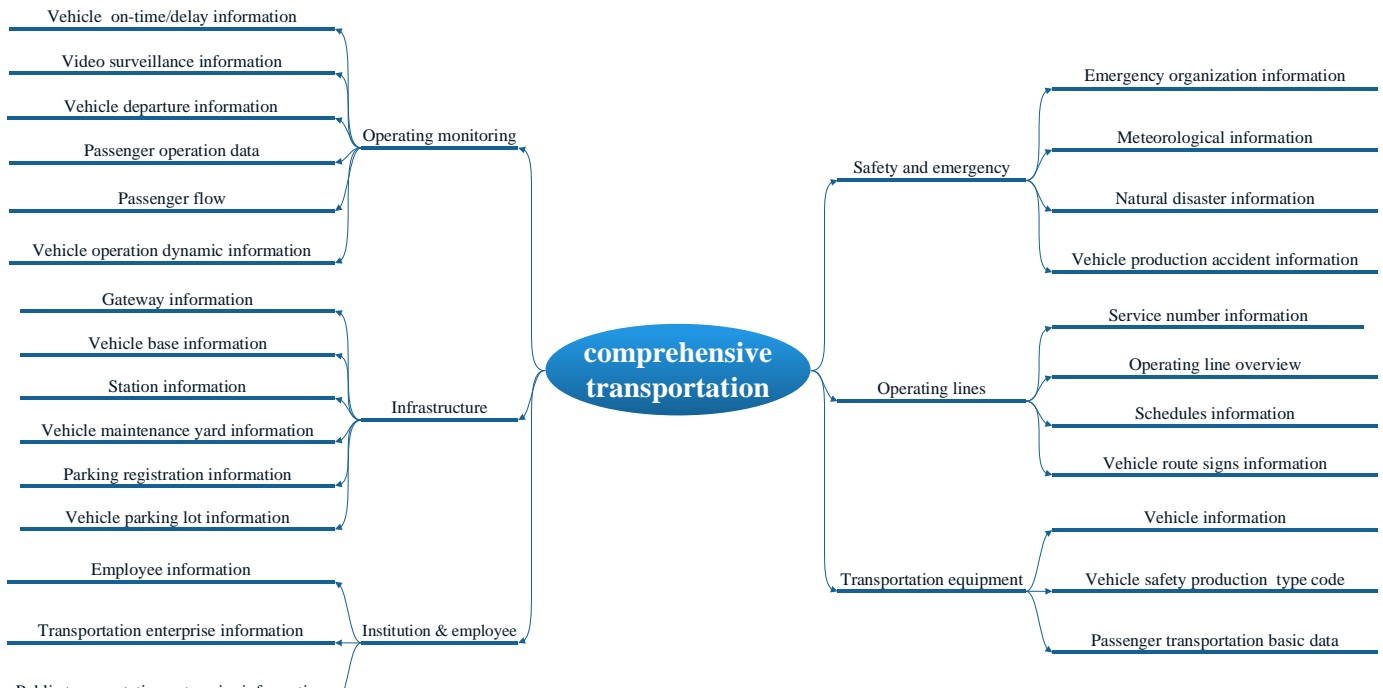

**Figure 1.** Summary of data exchange contents between different transportation systems.

*2.2. Comprehensive Assessment Framework for Data Trustworthiness in Intermodal Transportation*

Based on the characteristics analysis of the data above, we propose a comprehensive assessment framework for entity trustworthiness as shown in Figure 2, to meet the data exchange demand for cross-mode transportation collaboration. In this hierarchical framework, a single transportation mode is integrally divided into different network domains, based on main business classification. Further, every information system in the domain contains an indefinite number of datasets, which are defined by data safety hierarchy and content. Among the whole network, it is assumed that every entity in the information system layer has only one external interface to interact with both in-domain and out-of-

domain entities, so an information system is regarded as a minimum subject entity as well as data source in the access process.

According to the hierarchical structure, the assessment is composed of different aspects. Firstly, we propose a multi-demission model to evaluate data quality, in consideration of the reliability of data applications. And then, trust analysis network is initialized by decentralized data sources, thus data quality is taken as the most significant assessment criteria to initialize the trust network. The whole network topology varies as the data interactions carries out between entities. Meanwhile, the context interactions are introduced to calculate the trust inside and outside subject' domain. Finally, based on temporal knowledge graph, the dynamic changes in trust can be clearly displayed, which lays the foundation for subsequent intelligent applications, such as global security surveillance and a malicious behavior warning. The specific process is described in the following sections.

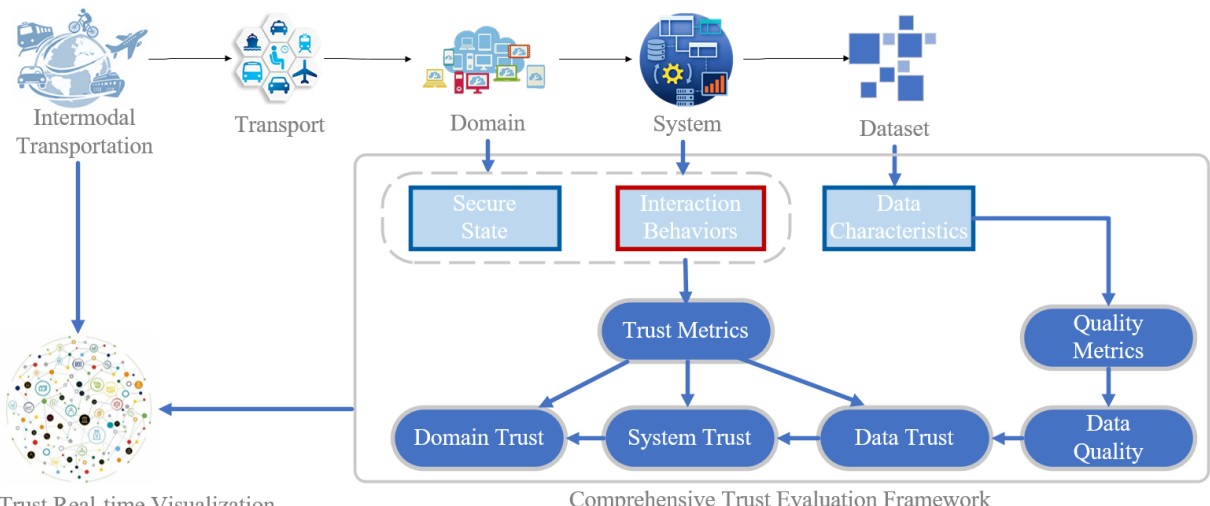

**Figure 2.** Comprehensive assessment framework for data interactions in intermodal transportation.

## 3. Quality Assessment for Collaborative Interaction of Transportation Data

Based on the exchange requirement and characteristics of transportation big data, this section selects appropriate dimensions for data quality assessment and specific metrics. Then, the corresponding evaluation models and detailed processes are defined. Here, the assessment method is a comprehensive assessment approach, including both a qualitative assessment and a quantitative assessment [7].

### 3.1. Metrics for Intermodal Transportation Data Quality Measurement

Since transportation big data exhibits distinct spatiotemporal characteristics, the temporal and spatial properties are as important as inherent content. Accordingly, we employ a three-dimensional structure, i.e., "Time + Space + Content", to evaluate the quality of transportation information resources, where the dimension of data quality refers to a characteristic or attribute of dataset that can be measured and improved. In fact, quality dimensions provide a way to measure and manage the quality of data products and information resources. Quality indicators for data product quality belong to these quality dimensions and represent a more detailed evaluation form of quality dimensions. Therefore, a two-layer assessment system is proposed with ten indicators from three dimensions, as stated in Table 1.

The time dimension contains three indicators: time coverage, timeliness, and stability. The space dimension contains two indicators: space coverage and spatial consistency. The content dimension contains five indicators: consistency, accuracy, integrity, validity, and traceability. The specific meanings of these ten indicators are defined in Table 1.

**Table 1.** Quality evaluation metrics for intermodal transportation data.

| Dimension | Quality Indicator | Definition |
|---|---|---|
| Time | Time Coverage | The completeness of data objects at various time points described in the dataset. |
| | Timeliness | The significant time difference from the generation to the acquisition and utilization of the data product. |
| | Stability | Measures the volatility and discreteness of the data with time. |
| Space | Space Coverage | The completeness of data objects included in the space described by the data product. |
| | Spatial Consistency | The correctness and completeness of logical relationships between different data objects in space. |
| Content | Consistency | The correctness and completeness of logical relationships between data objects in different attributes or between different objects. |
| | Accuracy | The degree of correctness, reliability, and distinguishability of the data product. |
| | Integrity | The completeness of the data product in terms of attribute sets. |
| | Validity | Describes whether the model or data satisfies user-defined conditions. |
| | Traceability | Whether the data product provides a description of the operations and transformations undergone during its lifecycle. |

### *3.2. Comprehensive Data Quality Assessment Model for intermodal Transportation*

With the evaluation indicators above, an evaluation system is proposed for intermodal transportation data interaction. The system consists of time quality, space quality and content quality. In specific application, these three parts can be used separately or jointly, according to demand of practical scenario. In addition, each quality value is limited to (0, 1), so as to ensure the generality of the model. For a dataset *P*, we assume that *P* has *N* data objects, i.e., $P = \{o_1, o_2, …, o_N\}$, and the objects make up an *LN*-layer spatial structure; *K* refers to the number of total timestamps of *P*, indicating time range; the attribute set of *P* is *A* = {$A_1$, $A_2$, …, $A_M$}, and *M* is the number of attributes.

### 3.2.1. Time Quality Evaluation

The model for time quality *PT* is as follows:

$$PT = w_{t1} \times PT_{COV} + w_{t2} \times PT_{TL} + w_{t3} \times PT_{STA} \tag{1}$$

where $w_{t1} \sim w_{t3}$ represent weights, limited by $w_{t1} + w_{t2} + w_{t3} = 1$, and the values can be determined by actual requirements or the importance of evaluation indicators. *PT$_{COV}$*, *PT$_{TL}$*, and *PT$_{STA}$* represent the evaluation results of time coverage, timeliness, and traceability, respectively.

(a) Time Coverage *PT$_{COV}$*

If data object $p_i$ is the lack of content at a certain time point, the time integrity of the data object is affected. Assuming the mapping function $F(x|C)$ represents whether the data object exists under certain conditions, we have:

$$F(x|C) = \begin{cases} 1, & data\ exists\ under\ condition\ C \\ 0, & else \end{cases} \tag{2}$$

Thus, the time coverage evaluation *PT$_{COV}$* of dataset *P*, is as follows:

$$PT_{COV} = \frac{\sum_{i=1,2,\dots,N; j=1,2,\dots,K} F(o_i | t = t_j)}{N \times K} \tag{3}$$

The value range of $PT_{COV}$ is (0, 1), where a value closer to 1 indicates a better time coverage of the data product, and vice versa.

(b)  Timeliness $PT_{TL}$

Timeliness reflects whether the generation of the data is timely, and it can be represented by the difference between the time of data generation and the current time. Assuming the current time is used as the reference time, denoted as $t$, the timeliness evaluation model $PT_{TL}$ is as follows:

$$PT_{TL} = 1 - \frac{t - t_P}{t} \tag{4}$$

where $t_P$ represents the creation or acquisition time of data $P$. For the ease of calculation, $t_P$ and $t$ can be converted into integers. For example, the time point $t_0$ can be uniformly converted to the time interval from 1 January 1970, 0:00:00 in milliseconds. The value range of $PT_{TL}$ is (0, 1), where a value closer to 1 indicates better timeliness of the dataset, and vice versa.

(c)  Stability $PT_{STA}$

Stability measures the volatility and discreteness of data. The dataset is divided into equal data slices based on the time range, and it is categorized into $N$ classes according to object characteristics. Calculate each expectation $\mu_i$ and standard deviation $\sigma_i$ of the newly added data within each class and obtain coefficient of variation $C \cdot V_i$ for each class. The overall dataset stability is defined as the average of the coefficients of variation for each class:

$$STA = \frac{1}{N} \Sigma \left( \left| \frac{\sigma_i}{\mu_i} \right| \right) \tag{5}$$

The range of $STA$ is (0, ∞), generally applicable when the average value is greater than 0. The smaller $STA$ is, the smaller the data fluctuation and the lower the degree of dispersion, indicating higher stability. To limit the range of $PV_{STA}$ to [0, 1], select the tanh function for mapping. Therefore, the stability evaluation model $PV_{STA}$ is:

$$PT_{STA} = \frac{2}{1 + e^{-2STA}} - 1 \tag{6}$$

3.2.2. Space Quality Evaluation

The evaluation model for space quality is as follows:

$$PS = w_{s1} \times PS_{COV} + w_{s2} \times PS_{CON} \tag{7}$$

where $w_{s1}$ and $w_{s2}$ represent weights, and $w_{s1} + w_{s2}$ = 1. The values of weights are determined by the application requirement. $PS_{COV}$ and $PS_{CON}$ represent the evaluation results of spatial coverage and spatial consistency, respectively.

(a)  Spatial Coverage $PS_{COV}$

Spatial coverage reflects whether data objects in the data product are missing or redundant. The spatial coverage evaluation model $PS_{COV}$ is as follows:

$$PS_{COV} = \begin{cases} \frac{count(P)}{N}, & 0 < count(P) \leq N \\ 1 - \frac{count(P) - N}{N}, & N < count(P) < 2N \\ 0, & count(P) \geq 2N \end{cases} \tag{8}$$

where the function *count*(*P*) refers to the amount of data objects in the dataset. If the value of $PS_{COV}$ is 1, it indicates that there are no missing or redundant data objects. The closer $PS_{COV}$ is to 1, the fewer missing or redundant data objects there are, and vice versa.

(b)　Spatial Consistency $PS_{CON}$

Since the way of obtaining data and the data standards both vary considerably among different transportation departments and organizations, the form and content of spatial data often have significant differences in geometric aspects. Therefore, in addition to spatial coverage, it is also necessary to check spatial consistency for traffic data.

Spatial consistency includes spatial location consistency, spatial objectives consistency, and spatial relationship consistency [6]. Spatial location consistency refers to the degree of matching in coordinate representation. Spatial objectives consistency refers to the equivalence of object existence, number of digits, shape, size, and spatial details. Spatial relationship consistency refers to topological equivalence and directional equivalence. Among these types of consistency, the relationship consistency is the most important content, with topological relationships dominating.

Assuming $o_{ik}$ and $o_{jl}$ represent the *k*th data object in the *i*th layer and the *l*th data object in the *j*th layer, $\eta_s(o_{ik}, o_{jl})$ is the spatial consistency check function between data objects $o_{ik}$ and $o_{jl}$. If the topological relationship between objects $o_{ik}$ and $o_{jl}$ is consistent with reality, $\eta(o_{ik}, o_{jl})$ is 1; otherwise, it is 0. The spatial consistency evaluation model $PS_{CON}$ is defined as follows:

$$PS_{CON} = \frac{\sum_{i,j=1,2,\dots,LN;k,l=1,2,\dots,N} \eta_s(o_{ik}, o_{jl})}{\sum_{i=1,2,\dots,LN,j=1,2,\dots,LN} tf_{ij}} \tag{9}$$

where $tf_{ij}$ represents the number of neighborhood object for $o_{ik}$ and $o_{jl}$ with topological relation, in the *i*th and *j*th layer.

### 3.2.3. Content Quality Evaluation

The evaluation model for content quality is as follows:

$$PV = w_{v1} \times PV_{COV} + w_{v2} \times PV_{ACC} + w_{v3} \times PV_{EFF} + w_{v4} \times PV_{CON} + w_{v5} \times PV_{TRA} \tag{10}$$

(a)　Attribute Coverage Rate $PV_{COV}$

If the attributes of data objects in the data product are missing, it will reduce the usability of the data product. The attribute coverage rate evaluation model $PV_{COV}$ is as follows:

$$PV_{COV} = \frac{\sum_{i=1,2,\dots,N;j=1,2,\dots,M} F(o_i|A = A_j)}{N \times M} \tag{11}$$

where $F(o_i|A = A_j)$ is to check the existence of the *i*th data object at the *j*th attribute. The $PV_{COV}$ value closer to 1 indicates a better time coverage of the data product, and vice versa.

(b)　Accuracy $PV_{ACC}$

Accuracy reflects whether data objects accurately and truthfully describe the application scenario. We suppose the attribute set $A = \{A_1, A_2, \dots, A_M\}$ has reference value standard sets $R = \{R_1, R_2, \dots, R_M\}$ in this case. Let $\varphi(\cdot)$ be the accuracy judgment function. If the value of object *oi* at attribute $A_k$ satisfies the reference value standard $R_k$, then $\varphi(o_{ik})$ is 1; otherwise, it is 0. The accuracy evaluation model $PV_{ACC}$ is as follows:

$$PV_{ACC} = \frac{\sum_{i=1,2,\dots,N;j=1,2,\dots,M} \varphi(o_{ij}|R_j)}{N \times M} \tag{12}$$

where the range of $PV_{ACC}$ is [0,1]. When $PV_{ACC}$ is 0, the accuracy of data objects is low; and vice versa.

(c)    Effectiveness $PV_{EFF}$

Effectiveness can be assessed by customized rules for specific application scenarios, where each rule can be related to one or more conditions. For example, when accessibility is the metric, we should inspect the physical conditions and interfaces for users to access data. The effectiveness evaluation model $PV_{EFF}$ is as follows:

$$PV_{EFF} = \frac{N - UN}{N} \tag{13}$$

where *UN* represents the number of ineffective data objects.

(d)    Consistency $PV_{CON}$

Consistency is used to determine whether the values between different attributes of the same data object are correct and complete. Let $A_k$ and $A_l$ be two attributes with consistency relations, and $\mu_v(\cdot)$ be the consistency judgment function. If the value of object $o_i$ at attributes $A_k$ and $A_l$ satisfies the consistency relation, then $\mu_v(o_{ik}, o_{jl})$ is 1; otherwise, it is 0. The consistency evaluation model $PV_{CON}$ is as follows:

$$PV_{CON} = \frac{\sum_{i=1,2,\dots,N;k,l=1,2,\dots,M} \mu_v(o_{ik}, o_{jl})}{N \times C_c(M)} \tag{14}$$

where the function *Cc(M)* counts the number of consistent attribute pairs in set *A*.

(e)    Traceability $PV_{TRA}$

The traceability evaluation is primarily qualitative [11]. It can be designed to score items that require traceability, and then check whether various traceable elements in the data are provided. If provided by the provider, the corresponding part of score is obtained; otherwise, the score for that item is 0. Finally, the obtained scores are added together to obtain the final evaluation result.

*3.3. Assessment Process*

Since transportation data covers all aspects of the transportation industry, the evaluation indicators, weights, and results threshold in the transportation data assessment model should be decided by the specific business requirement. In practical applications, after determining the need for data quality assessment, the evaluation of transportation data quality can be divided into the following steps:

(a)    Identify the evaluation dataset: select either the entire data content or a typical data field set as the evaluation object based on business needs.

(b)    Select evaluation indicators: determine the indicators or factors that need to be assessed. The choice of evaluation dimensions can be comprehensive, or to select individual dimensions based on business needs.

(c)    Determine evaluation weights: determine the factors influencing weight allocation, such as data importance, business requirements, expert opinions, etc., and select suitable weighting methods. Common methods for determining weights include the Analytic Hierarchy Process, Delphi method, statistical analysis method, and expert scoring method, etc., which should be chosen according to the content of data evaluation [12]. Continuously improve weight allocation based on feedback and actual results during the evaluation process to ensure the accuracy and effectiveness of the evaluation system.

(d)    Validate and Adjust: Verify whether the chosen indictors and calculated weights align with reality and make adjustments and optimizations as needed. Relevant data and stakeholders' opinions should be gathered to provide a basis for validation.

(e)    Summarize: record the evaluation results of each dataset, calculate the score of the evaluation object based on the weights and scores of each dataset, and determine whether the data is compliant based on the threshold.

## 4. Trustworthiness Evaluation for Data Collaborative Interaction of Intermodal Transportation

In the data access session, trustworthiness reflects the degree to which the subject entity meets the object security expectations. It varies with every access behavior. Thus, it is characterized by subjectivity, context dependency, dynamic uncertainty, and temporal lag. Considering the cross-domain interactions of traffic information resources, a hierarchical framework is proposed to inspect entity trustworthiness on different levels. By refining level by level, the evaluation system transforms the trust measurement problem of complex network behavior into an objective, measurable and computable evaluation problem, based on interaction evidence.

### 4.1. Trustworthiness Assessment Framework for Data Interaction of Intermodal Transportation

#### 4.1.1. Hierarchical Structure

Based on the cross-mode interactions of transportation information resources, a hierarchical entity trust assessment model is proposed, as shown in Figure 3. This framework facilitates a hierarchical approach to comprehensively assess based on the mixed evidence layer, making the evaluation of complex network behaviors more objective and quantifiable. The calculation of trustworthiness consists of three layers: data layer, information system layer, and secure domain layer.

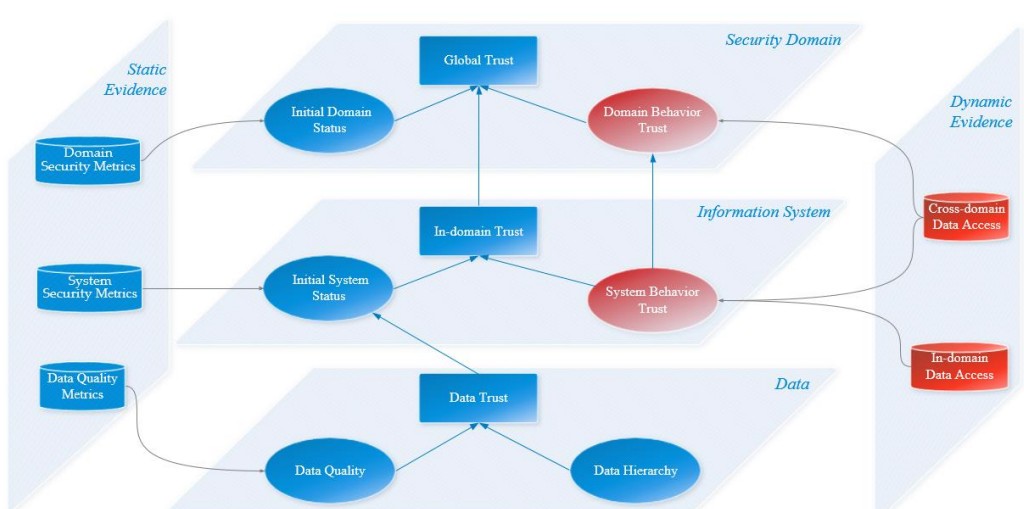

**Figure 3.** Illustration for hierarchical entity trustworthiness assessment framework.

Aside from the multi-level structure, this framework is characterized by the multi-part evidence layer. As for the cross-mode interaction of traffic data, trustworthiness is dependent on both the intrinsic properties of the subject entity and its past performances. In order to evaluate the trust of data and entities comprehensively, the measurement of trustworthiness should consist of two parts, i.e., the static trust indicates entity secure state while dynamic trust indicates entity historical access behaviors. Generally, the static trust and the dynamic trust can complement each other, and dynamic trust provides more fine-grained security than the static identity trust. Considering the actual situation of the communication network compared with the frequent data access between systems, the inherent characteristics of the system and security domain are relatively fixed. Therefore, in order to simplify the modeling and calculation, we assume that the static trust is kept in a relatively stable state from beginning, so it is expressed by the initial state. When the entity suffers from network attack or malfunction, which leads to sharp changes in inherent state, it is necessary to reinitialize the evaluation network according to the situation at that

moment. Therefore, this framework is based on an abundant evidence layer composed of a static part and dynamic part.

As for different types of evidence, they are dealt with corresponding metrics. For static trust, different evaluation metrics can be used for entity types at different levels, from the perspective of actual security requirements in practical application. Depending on practical considerations, additional evidence types can be introduced, or existing ones may be adjusted to balance feasibility and computational complexity in evidence collection and handling. For dynamic trust, the access success rate is the key indicator.

### 4.1.2. Operating Principle

When the model is initialized, the self-trustworthiness of the requesting domain is set to the initial trustworthiness, and basic network access control rights are allocated based on trustworthiness. The interaction context is continuously monitored and measured. If at time $t$ there is no new evidence of interaction context compared to time $t - \Delta t$, the direct trustworthiness at the current time decays over time. When the direct trustworthiness increment is positive, no punishment is required. However, when the increment is negative, a penalty is applied to the trustworthiness to make it quickly drop below the trust threshold, restricting its access control rights.

During the process of updating and iteration of the trust calculation network, regulation of historical behavior and malicious access behaviors is of great significance [13]. Thus, we introduce the sequence factor, temporal factor and penalty factor to adjust the dynamic changes. The sequence factor inspects historical behaviors, the temporal parameter targets the attenuation with time, and the penalty factor deals with impact of malicious access.

(a) Sequence factor

The impact of an entity's historical access behavior on the current trust value varies at different moments. Typically, access behaviors closer to the present have a greater impact on the entity's trust value, while those that occurred earlier have a smaller impact. Therefore, a sequence decay factor $H(k)$ is introduced, to adjust the influence of sequential access behaviors, expressed as:

$$H(k) = \frac{k}{K}\rho, \quad k = 1, 2, \cdots, K \tag{15}$$

where $k$ is the sequence number of access request, and $K$ is the count of total historical access behaviors. $H(k) \in (0, 1]$, $\rho \in (0, 1]$, $\rho$ is used to adjust the decay rate, with a larger value leading to faster trustworthiness decay. The sequence factor takes into account the dynamic decay of trustworthiness over access requirements, effectively enhancing the accuracy of trustworthiness calculation.

(b) Temporal factor

Temporal factor represents the time attenuation coefficient at time $t$ [14]. When the trust of entity remains the same from time $t - \Delta t$ to $t$, it is subjected to a time-attenuation penalty. Temporal factor is defined as:

$$\lambda(t) = 1 - \frac{\Delta t \times \xi}{t - t_0} \tag{16}$$

where $\lambda(t) \in (0, 1)$, $\xi \in (0, 1]$, $t_0$ is the starting time for calculation, $t$ is the current time, and $\Delta t$ is the time difference between two consecutive calculations. $\xi$ is used to adjust the attenuation rate, with a larger value leading to faster trustworthiness decay. The temporal factor takes the dynamic decay of trustworthiness over time into account, effectively enhancing the accuracy of trustworthiness calculation.

(c) Penalty factor

For undesirable behaviors in entity interaction, a penalty factor $\delta$ is introduced, expressed as:

$$\delta = \begin{cases} 1, & TD(t) - TD(t-1) \geq 0 \\ 0 < \delta_t < 1, & TD(t) - TD(t-1) < 0 \end{cases} \tag{17}$$

where $\delta t \in (0, 1)$, and *TD* is the behavioral trust. When *TD* decreases at *t*, a penalty is applied to the trust, and $\delta_t$ is used to adjust the strength of the penalty. A smaller value of $\delta_t$ results in a stronger penalty, while a larger value leads to a milder penalty. The penalty factor is crucial for imposing strict punishment on entities providing false or malicious services, causing them to rapidly decrease below the trust threshold and restricting their access control permissions in the whole network. This effectively curbs the attacks of malicious entities.

*4.2. Hierarchical Data Trust Assessment for Data Interaction*

4.2.1. Data Trust

The trustworthiness of dataset *d* is determined by the inherent characteristics of the data. Aside from data quality, the grade is also of great importance since it is based on the universal standard dedicated to importance and sensitivity. Data trust is represented as $T_{data} = \sum_{s \in P_{data}} w_s e_s$, where $e_s$ represents the trust value determined according to the requirement in $P_{data}$, and $w_s$ represents the corresponding weight. Default $w_s$ and $e_s$ are shown in Table 2.

**Table 2.** Requirement of security for data.

| Data Security Requirements $P_{data}$ | Weight $w_s$ | Evaluation Value $e_s$ |
|---|---|---|
| Time quality | 0.2 | *PT* |
| Space quality | 0.2 | *PS* |
| Content quality | 0.2 | *PV* |
| Data grade | 0.4 | Five levels {0.2, 0.4, 0.6, 0.8, 1} |

4.2.2. In-Domain Trust

In-domain trust describes the credibility of a system, from the perspective of other systems in the same secure domain, based on its initial state and all past interaction requests.

At the beginning, the initial trust of the system *u* is determined by its own security attributes, including the data-layer and system-layer information. In-domain trust of *u* is expressed as $T^u(0) = \sum_{s \in P_{system}} w_s e_s$, where $e_s$ represents the value determined by the security requirements of its own domain $D_i$. The security requirements and weights for *u* are shown in Table 3.

**Table 3.** Initial requirement of security for system.

| System Security Requirements $P_{system}$ | Weight $w_s$ | Assessment Value $e_s$ |
|---|---|---|
| The identity information integrity | 0.2 | In case of complete1; otherwise 0. |
| The system secure grade | 0.3 | Five levels {0.2, 0.4, 0.6, 0.8, 1} |
| The average quality of data provided by *u* | 0.4 | $\sum_{s=1} a_s Q_s(s)$ |

Within time interval $(t - \Delta t, t)$, the dynamic trust of system *u* can be obtained from the access behaviors during this period and is expressed as:

$$DTD_u(t - \Delta t, t) = \frac{\sum_{s=1}^{n} a_s H(s)}{\sum_{s=1}^{n} a_s H(s) + \sum_{s=1}^{m1} b_s H(s) + \sum_{s=1}^{m2} c_s H(s)} \tag{18}$$

where *n* is the number of successful access requests, while $m_1$ and $m_2$ represent the number of failed ones within the entity's domain and out of this domain, respectively. $a_s$ is the reward coefficient for successful requests, and $a_s \geq 1$. $b_s$ and $c_s$ are the punishment coefficients for domain access failures and cross-domain access failures, respectively, with

larger punishment coefficients indicating stronger penalties. Typically, the results of cross-domain access also affect the trust level between domains; therefore, the punishment for cross-domain access failures should be greater than that for domain access failures, so $c_s > b_s > 0$.

Therefore, the trust of the access system $u$ in its own domain $D_i$ is expressed as:

$$T^u(t) = \begin{cases} T^u(0), & t = 0 \\ T^u(t - \Delta t) \cdot \lambda(t), & \Delta N_{t-\Delta t,t} = 0 \\ \left(1 - \dfrac{\Delta t}{\alpha + t}\right) T^u(t - \Delta t) \cdot \delta + \dfrac{\Delta t}{\alpha + t} DTD_u(t), & \text{else} \end{cases} \tag{19}$$

where $\Delta N_{t-\Delta t,t} = n + m_1 + m_2$ represents the total number of access requests from $u$ during time interval $(t - \Delta t, t)$, $\alpha$ is the coefficient for historical trust value. $\lambda(t)$ represents the timeliness factor, and $\delta$ represents the penalty factor.

### 4.2.3. Global Trust

Global trust refers to the credibility of a system, from the perspective of systems in a particular secure domain, based on the initial state of requesting domain, in-domain trust and cross-domain trust, which measures all past interaction requests between these two domains.

At the initial moment, the trust of the requesting domain $D_i$ is determined based on environmental requirements and is expressed as: $T_i(0) = \sum_{s \in P_{domain}} w_s e_s$, where $e_s$ represents the trust value determined by the security requirements of the target domain $D_j$, and $w_s$ is the weight. The default metrics and weights of the target domain for the requesting domain are shown in Table 4.

**Table 4.** Initial requirement of security for domain.

| Domain Security Requirements $P_{domain}$ | Weight $w_s$ | Evaluation $e_s$ |
|---|---|---|
| Integrity of system authentication requirements within $D_i$ | 0.4 | In case of complete1; otherwise 0. |
| Integrity of $D_i$ after initializing | 0.3 | In case of complete1; otherwise 0. |
| Integrity of historical access behavior log within $D_i$ | 0.1 | In case of complete1; otherwise 0. |
| Integrity of the trust management point within $D_i$ | 0.1 | In case of complete1; otherwise 0. |
| Integrity of the policy information point in $D_i$ | 0.1 | In case of complete1; otherwise 0. |

Within the time range $(t - \Delta t, t)$, the dynamic cross-domain trust of the behavior of the requesting domain $D_i$ in the target domain $D_j$ is expressed as:

$$ITD_{ij}(t - \Delta t, t) = \frac{\sum_{s=1}^{N} a_s H(s)}{\sum_{s=1}^{N} a_s H(s) + \sum_{s=1}^{M} c_s H(s)} \tag{20}$$

where $N$ is the total number of successful access requests from any systems in domain $D_i$ to target domain $D_j$, $M$ is the total number of failed ones.

Therefore, the comprehensive trust from requesting domain $D_i$ to target domain $D_j$ is represented as:

$$T_{ij}(t) = \begin{cases} T_i(0), & t = 0 \\ T_{ij}(t - \Delta t) \cdot \lambda(t), & \Delta M_{t-\Delta t,t} = 0 \\ \left(1 - \dfrac{\Delta t}{\beta + t}\right) T_{ij}(t - \Delta t) \cdot \delta + \dfrac{\Delta t}{\beta + t} ITD_{ij}(t), & \text{else} \end{cases} \tag{21}$$

where $\Delta M_{t-\Delta t,t} = N + M$ represents the total number of access request from $D_i$ to $D_j$ in the time period $(t - \Delta t, t)$, $\beta$ is the coefficient for historical trust value. $\lambda(t)$ represents the timeliness factor, and $\delta$ represents the penalty factor.

In general, the global trust of entity $u$ in the requesting domain $D_i$ to the target domain $D_j$ is given by:

$$T_{ij}^{u}(t) = T_{ij}(t) \times T^{u}(t) \tag{22}$$

## 5. Data Interactive Visualization Method Based on Temporal Knowledge Graphs

Through graph visualization, suitable data display layouts and interactive methods can be designed according to the need of an intermodal transportation business. It helps information resource managers to perceive global data resources and understand cross-domain interactions. Therefore, a method based on a temporal knowledge graph is raised, and it helps to present massive and complex data interaction relationships intuitively and reasonably. In further research, it will lay the foundation of discovering malicious entities and tracing data flows promptly [15,16].

In this section, the Data Interaction Temporal Knowledge Graph (DITKG) is constructed for transportation cooperation. We first make the definition of DITKG, then introduce the methods for constructing the graph, and finally visualize the DITKG using the Neo4j graph database.

### 5.1. Formal Definition of DITKG

Knowledge graphs have evolved from the development of graph data technologies. In contrast to traditional tabular storage methods, the knowledge graph data model is more suitable for machines to understand data correlations, aligning with human cognition and memory of the real world. As data volumes grow and business complexities increase, traditional forms of charts and metrics may not satisfy the need for business staff to understand the relationships behind the results. Furthermore, time information is significant to represent the variation trend of domain trustworthiness between two security domains, and situations where the global trustworthiness of a system sharply declines due to continuous malicious behavior, etc. Furthermore, it can provide vital support in the analysis of trustworthiness, predicting cross-domain malicious access, and thus protecting the security of data interactions. Therefore, we proposed a visualization approach based on temporal knowledge graph.

The data interaction temporal knowledge graph is a heterogeneous graph that describes interaction relationships between entities, where access requests are represented by directed edges with timestamps. Each fact in the temporal knowledge graph can be represented by a temporal quadruple (*es*, *r*, *eo*, *t*), where *es* is the head entity, *r* is the relationship between the head entity and the tail entity, *eo* is the tail entity, and *t* represents the timestamp of the occurrence. This quadruple overall describes the interaction relationship *r* occurring between the subject entity *es* and the object entity *eo* at time *t*. In addition, the set of all content in the DITKG is defined as *G* = (*E*, *R*, *T*), where *es*, *eo* ∈ *E* represents the entity set, *r* ∈ *R* represents the relationship set, and *t* ∈ *T* represents the timestamp set.

### 5.2. Schema of Temporal Data Knowledge Graph

The formal definition of the temporal knowledge graph lays the foundation for constructing the graph. The detailed construction involves ontologies, relationships, and attributes of the DITKG.

Since the constructed knowledge graph in this paper is time-varying, there are both time-sensitive and time-insensitive attributes. For example, attributes such as in-domain trust and access count are time-sensitive, closely linked to time, and their values may change at different moments. On the other hand, attributes like data ID and the partition of security domains are time-insensitive; they do not change with time and have fixed

values. As shown in Figure 3, dynamic attributes are all related to the interaction behaviors, which are presented as relationships in DITKG. Therefore, attributes of ontologies are time-sensitive while that of relationships are time-insensitive.

A.   Entity and Attribute Definitions

According to the trust assessment framework, ontologies in this graph include data sets, information systems, security domains and transport mode. Ontologies with their attribute as defined in Table 5, which describes the static attributes corresponding to each ontology.

**Table 5.** Temporal knowledge  graph ontology and attribute definition.

| Entity | Attribute | Description |
|---|---|---|
| Data | Data_ID | Dataset ID |
| | Data_name | Dataset name |
| | Data_quality | Quality of a dataset |
| | Data_state | Initial state of a dataset |
| System | Sys_ID | System ID |
| | Sys_name | System name |
| | Sys_state | Initial state of an information system |
| Domain | Domn_ID | Domain ID |
| | Domn_name | Domain name |
| | Domn_state | Initial state of a security domain |
| Transport | Transport_ID | ID of a transport mode |
| | Transport_name | Name of a transport mode |

B.   Relationship and Attribute Definitions

Relationships in the DITKG consist of two types: memberships and interactions. Memberships are static without time-varying attributes, referring to the relation "is an element of". Memberships exist between a system and dataset, a system and a security domain, a security and a transport mode. Interactions describe the access request between subject entity and object entity, and they are the most significant part in DITKG with time-sensitive properties. From the perspective of accessing entity types, interactions are defined between System and Data, System and System, System and Domain, Domain and Domain. To reflect the changes in trust value adequately, interactions between System and Domain are divided into System and Domain to which it belongs, and System and other Domain, corresponding to the in-domain trust and global trust, respectively.

In the process of generating the temporal quadruple knowledge graph, there are two ways to represent time information [13]. One is as a new relationship connecting the subject entity and the object entity, and the other is as an attribute of the relationship itself. In this paper, the treatment of time-sensitive attributes is consistent with the approach used in relation extraction, considering this type of data as four-tuples with time labels [17]. Table 6 describes the relationships and corresponding associated attributes.

**Table 6.** Relationships and corresponding associated attributes of knowledge graph.

| Relationship | Subject | Object | Attribute | Description |
|---|---|---|---|---|
| con_data | System | Data | - | - |
| con_sys | Domain | System | - | - |
| con_domn | Transport | Domain | - | - |
| access_data | System | Data | result | The latest result of access to object by subject in Boolean. |
| | | | count | The total number of access requests by subject to object. |
| access_sys | System | System | sucs_rate | The probability of access success among all requests by subject to any dataset of the object system. |

| | | | count | The total number of access requests to any dataset of object system. |
|---|---|---|---|---|
| access_indomn | System | Domain | indomn_trust | The value of in-domain trust. |
| | | | count | The total number of in-domain access requests by subject. |
| access_outdomn | System | Domain | global_trust | The value of global trust. |
| | | | count | The total number of out-of-domain access requests by subject. |
| inter_domn | Domain | Domain | inter_trust | The inter-domain trust from subject to object. |
| | | | count | The total number of cross-domain access requests by systems in subject domain. |

In summary, the constructed pattern layer is illustrated in Figure 4. As shown, static properties and relationships are represented in blue, while dynamic ones are in red.

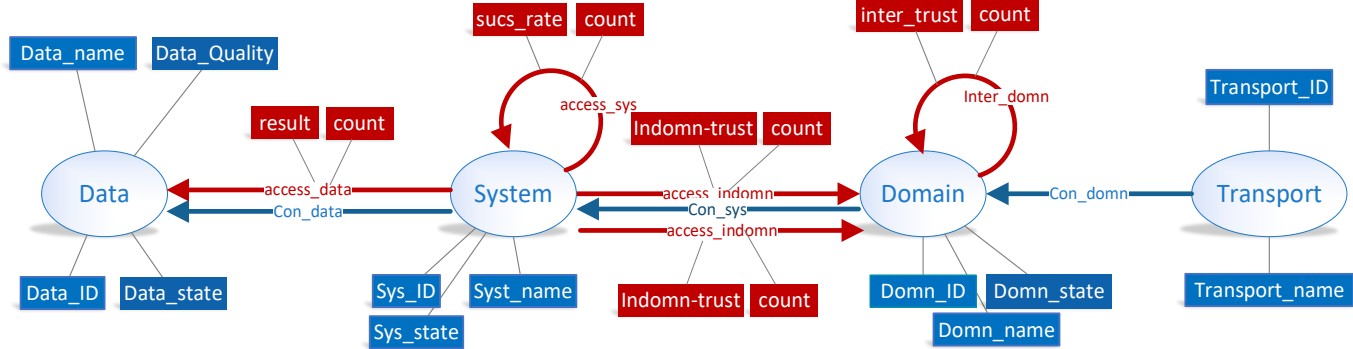

**Figure 4.** Schema of temporal knowledge graph for data interaction in intermodal transportation.

*5.3. Construction and Visualization of Temporal Knowledge Graph*

Graph visualization involves transforming the originally obtained relational database into a graph database schema based on the schema illustration. Compared to common knowledge graph, the DITKG extends triples into temporal quadruples (*es*, *r*, *eo*, *t*) to take dynamic temporal information into account, where *t* provides additional temporal information about when the event, i.e., an access request occurred.

We use the Neo4j graph database to draw the temporal knowledge graph of data interactions. As a graph database, Neo4j stores knowledge in a network form of ontology structure, unlike traditional relational databases that use tabular forms [18]. It visualizes relationships between entities and is one of the most commonly used databases for knowledge graphs. The triple knowledge of Neo4j is usually stored in the "node-edge-node" format, while temporal quadruples need to be stored in DITKG. Therefore, we added the timestamp as an attribute of the relationship in Neo4j when storing. For example, the quadruple ('AFC', 'access_data', 'Railway Ticket Reservation Data', '2023/10/19/13/0/0') would be stored as an 'access_data' relationship with the time attribute '2023/10/19/13/0/0'. Finally, to improve the intelligibility of the graph, we stipulate that the ontology of the graph is displayed as its own name attribute by default, the static relationships are displayed as types, while the dynamic relationships are displayed as the trust attribute.

## 6. Application and Analysis

This section verifies the comprehensive trust assessment framework for information resource exchange in intermodal transportation. Based on the scenario of intermodal passenger transport between subway and mainline railway, we first calculate the quality of relevant datasets, to check whether it can meet the demand of the cross-mode standard. And then, the trust of entities is compared in different situations. Finally, the dynamic changes caused by data interaction is visualized by DITKG.

For intermodal passenger transport, the demand of information exchanging mainly focuses on one-ticket booking [19], intelligent passenger guide, and better transfer support at junction station [20]. Some of related entities in subway and mainline railway are shown in Figure 5. In the subway system [21], the production management system (PMS) and production auxiliary systems (PAS) are regarded as the domain of subway operation business [22]. PAS has two significant information systems, i.e., passenger information system (PIS) and auto fare collection (AFC) system. The PIS system utilizes operational information that would be open to the public, through displays at stations, onboard broadcasting and displays, mobile applications, and online websites. The ticketing internet platform (TIP) is to update ticket availability, prices, and seat selection information to purchase tickets. AFC collects data from turnstiles and ticket selling equipment, and aggregates them to a central server, including passenger flow for income clearing, origin–destination passenger flow, cross section passenger flow, and ticket transaction data [23]. In railway system [24], the information systems associated with passenger service are the ticket reservation system (TRS), passenger transport marketing decision-support system (PTDS), passenger service systems (PSS), and passenger transport management system (PTMS). The first three of these belong to the business domain of passenger transport marketing (PTM), and the last one belongs to the transportation production organization (TPO) domain [25].

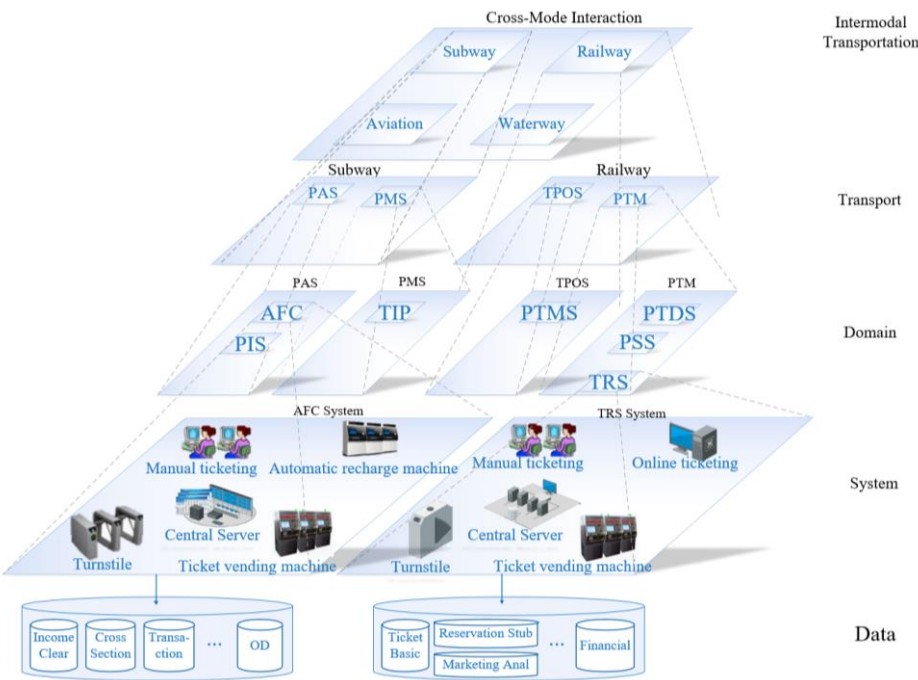

**Figure 5.** Part of related entities to intermodal passenger services in subway and railway.

### 6.1. Compliance of Data Quality

Taking a two-day ticket transaction dataset as an example, we conduct a quality assessment of the data and use 0.75 as the threshold to check the compliance of the dataset. This dataset is adopted from AFC system of subway transport in city A, where there are

a total of 11 subway lines in the city. In this case, the dataset is divided based on the line numbers where the passenger alighted, so that we can compare the data quality between different operating lines.

Firstly, the data feature analysis of subway ticket transaction data is conducted. Due to the fixedness of operating lines and stations, the spatial consistency is always 1. Also, because the data within the subway system is traceable from collection to aggregation, its traceability is considered as 1. The acquisition frequency of ticket transaction data is daily, and the dataset meets the requirements, hence its timeliness is 1. Therefore, the dataset's quality is primarily examined from metrics such as spatial integrity, temporal integrity, attribute completeness, validity, accuracy, stability, and consistency. The result is shown in Figure 6.

As in Figure 6, it can be observed that the stability of data from various subway lines is generally at a low level, where the time slice for stability evaluation is set to 1 hour. This is determined by the traffic patterns in urban transportation, and it varies dramatically from hour to hour. Assuming weights of quality metrics in Section 3.2 are determined by expert scoring method to be $w_{vi} = 0.2(i = 1,2,...,5)$, $w_{t1} = 0.6$, $w_{s1} = 0.8$, $w_S = 0.3$, $w_T = 0.3$, $w_V = 0.4$, the data quality assessment values for each line can be obtained. Line 10 has the highest value at 0.798, while Line 1 has the lowest value at 0.7432. The data of Line 1 are below the threshold of 0.75, indicating a need for improvement before sharing to other transport modes. The rest of the lines meet the standards.

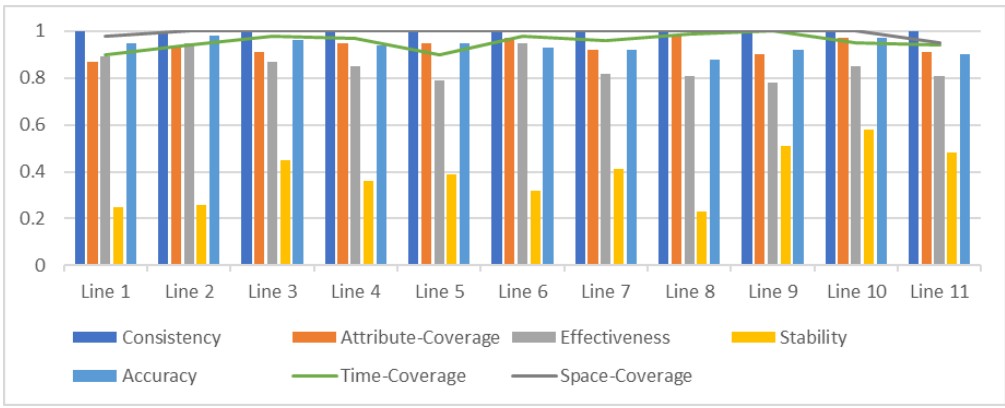

**Figure 6.** Quality of transaction data from 11 subway lines.

### 6.2. Trust Evaluation

In this experiment, we assume that the initial in-domain trust is set to 1 for all systems, and the initial inter-domain trust is also set to 1. The reward coefficient $a_s = 1$, the in-domain punishment coefficient $b_s = 1.5$, the inter-domain punishment coefficient $c_s = 2$; $\alpha = \beta = 1$; temporal factor $\lambda = 0.999$, and penalty $\delta = 0.98$. The highest frequency of access requests is once per second.

#### 6.2.1. Trust of a Subject System to Multiple Object

We assume that in the PAS domain, only AFC can raise cross-domain access request. The AFC system initiates access requests to the PIS server for the first 250 s, with 150 successes and 100 failures. From 251 to 500 s, access requests are made to the TRS in PTM domain. The changes in the in-domain trust and global trust of AFC system, as well as inter-domain trust between PAS and PTM over time, are shown in Figure 7.

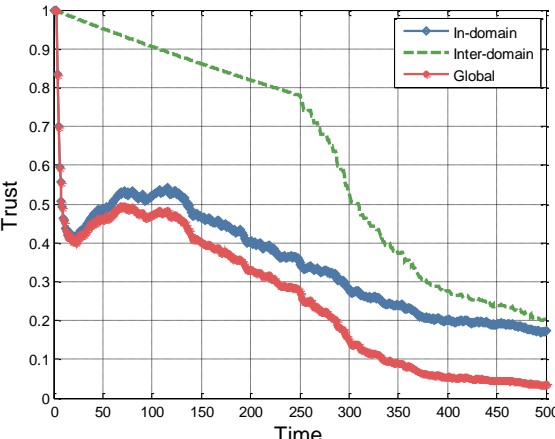

**Figure 7.** Performance of in-domain trust and global trust to PTM domain during 500s of mixed in-domain and inter-domain interaction.

Figure 7 illustrates the trust value curve of the AFC system under random success and failure conditions. The experimental results show that the entity's trust value increases with the count of successful accesses and decreases with the count of failures. Additionally, under the influence of the penalty factor, the overall trend of the curve is downward. In the first 250 s, because there are no inter-domain access requests, the inter-domain trust decays over time due to the temporal factor. Therefore, the difference between the in-domain trust value of the AFC system and the global trust is relatively small. However, in the subsequent 250 s, as cross-domain interactions occur, the inter-domain trust fluctuates greatly, resulting in a gradual widening of the gap between the in-domain trust value and the global trust value.

Figure 8 shows the change in DITKG over time, where the relation "access_data" and trust-related relationships are in orange and red, respectively. For the ease of observation, only relationships in connection with the last interaction happened at the specific moment is visible. For example, AFC accesses the terminal data of PIS at *t* = 100 s and *t* = 200 s, so the graphs in (a) and (b) have the same structure with different transient attributes. In this scenario, graphs in (c) and (b) are completely unlike.

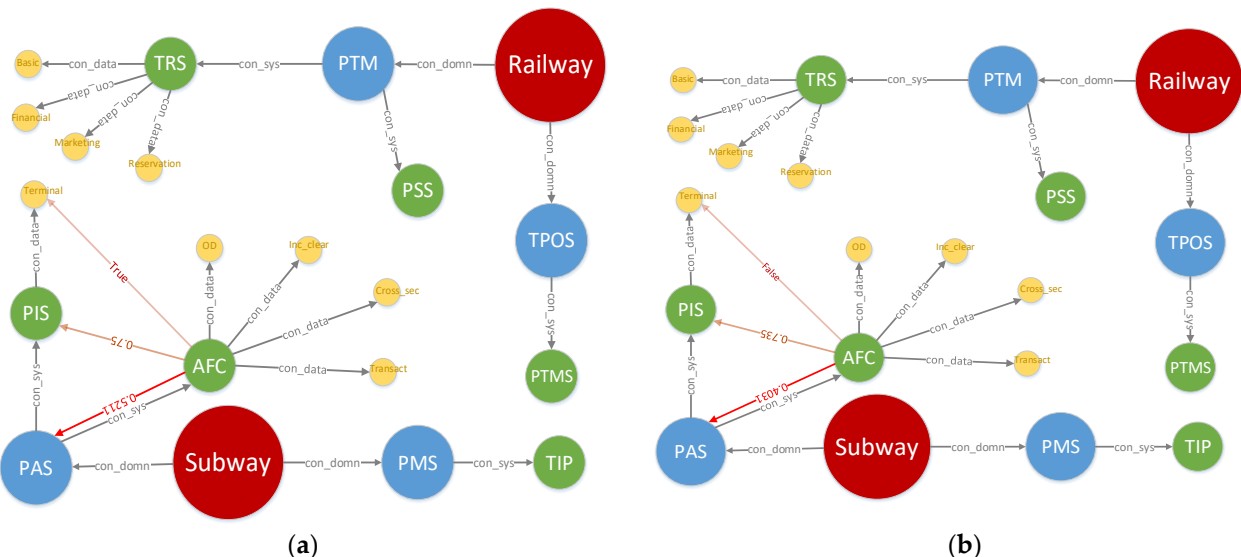

(**a**)　　　　　　　　　　　　　　　　(**b**)

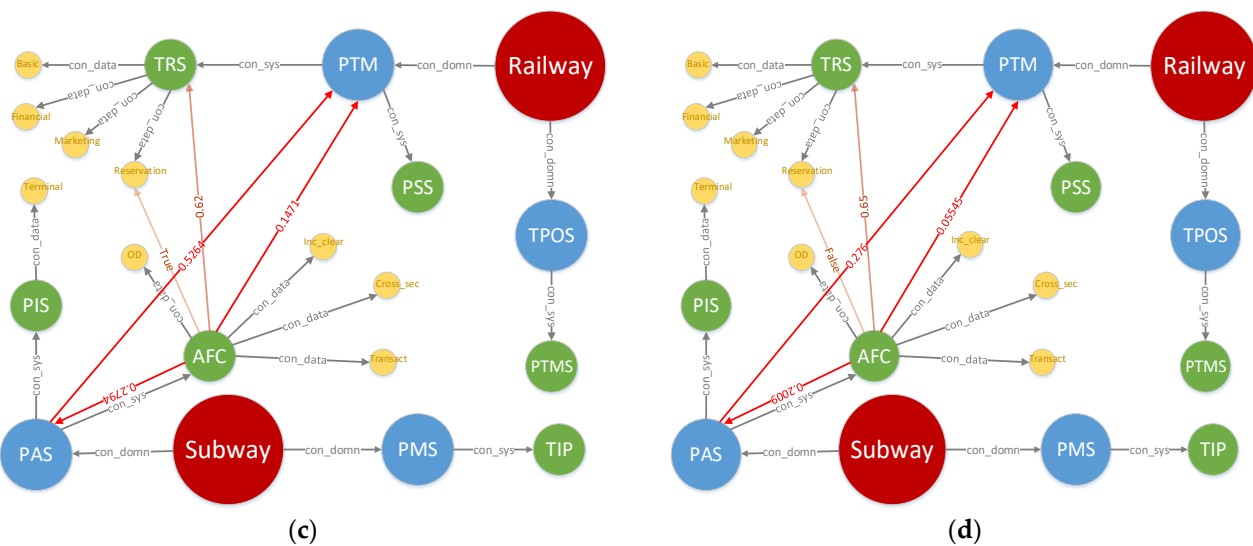

**Figure 8.** Visualization of DITKG in different time: (**a**) *t* = 100 s; (**b**) *t* = 200 s; (**c**) *t* = 300 s; (**d**) *t* = 400 s.

### 6.2.2. Mutual Effect of Multiple In-Domain Systems

To facilitate observation, the history records of access failures and successes are artificially controlled in this case. The AFC system and the PIS system both launch service access requests 500 times to a fixed object, and they may access each other or turn to the TRS in PTM domain. The AFC system fails in request 51 to 150, while the PIS system fails in 201 to 300, and other accesses are successful. The in-domain trust and global trust of both systems are shown in Figure 9, as well as inter-domain trust from PAS to PTM.

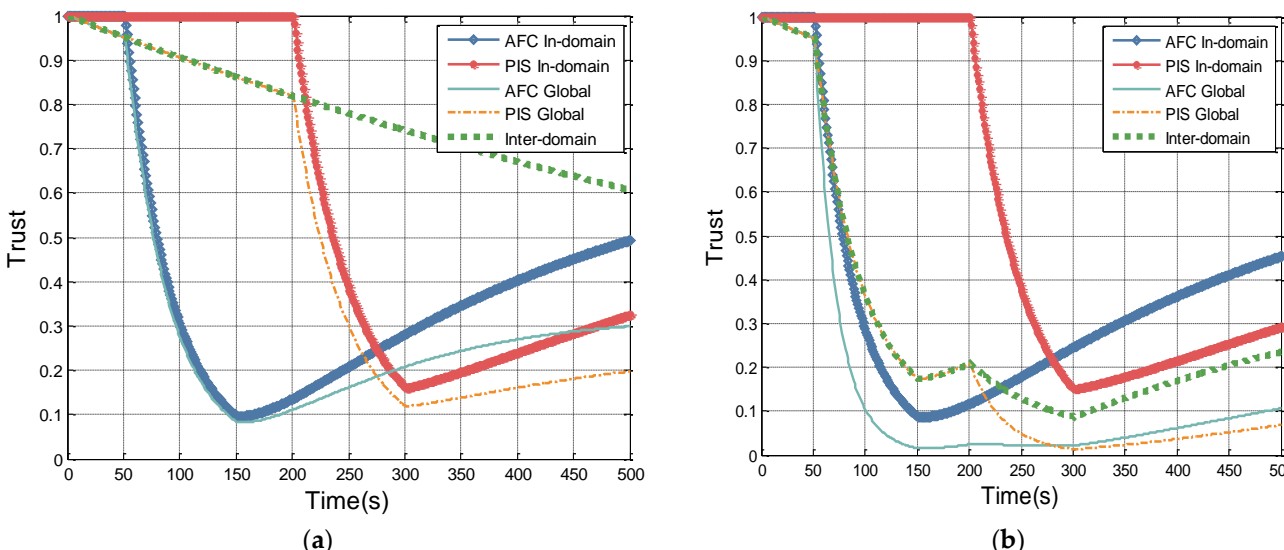

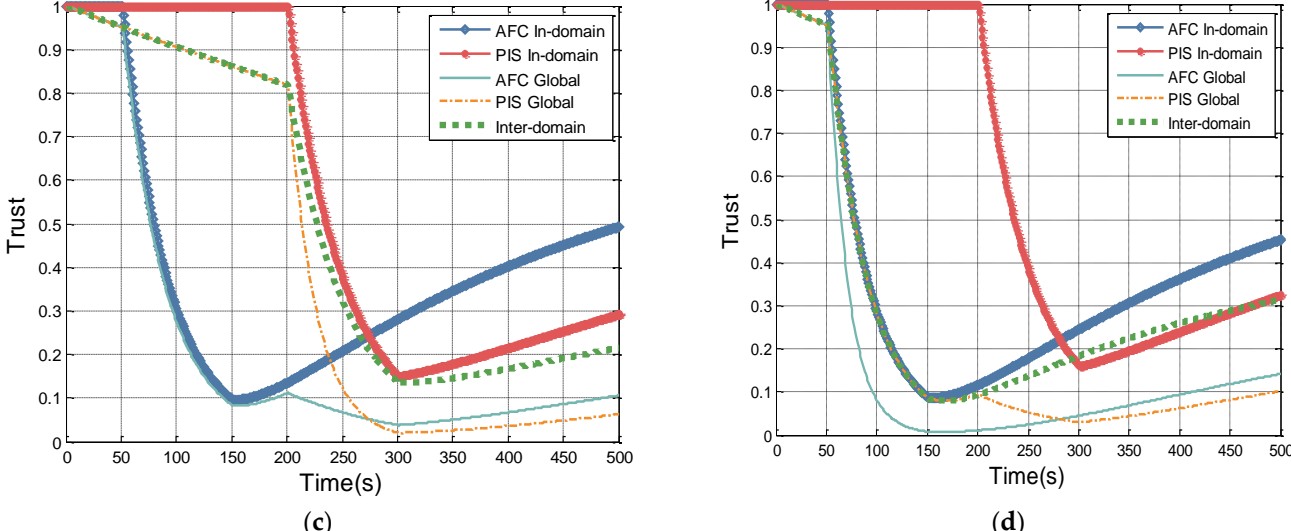

(**c**)                                                       (**d**)

**Figure 9.** Performance of in-domain and global trust of 2 systems in different scenarios: (**a**) AFC and PIS both access in-domain data; (**b**) AFC and PIS both access PTM domain data; (**c**) AFC accesses in-domain data while PIS accesses PTM domain data; (**d**) PIS accesses in-domain data while AFC accesses PTM domain data.

Figure 9 illustrates the trust value changes in cases where access objects are different. Firstly, let us examine the case where no cross-domain accesses occur, as shown in (a). As the inter-domain trust value only decays over time, the only influence factor is the access result sequence. Observing the in-domain trust curves of AFC and PIS in (a), it can be seen that the lowest point of the AFC curve is lower than that of the PIS curve. This is because the records of access failures of the AFC system are relatively distant from the current moment compared to PIS, resulting in a higher proportion of access failures relative to the total number of accesses at the specific time. Additionally, at 500 s, the trust of AFC is higher than that of PIS, as the impact of access history records on the trust value of an entity increases with proximity to the current time point.

Furthermore, considering the cases where cross-domain interactions happen, when we compare the results in (d) and (c), the fluctuation trend of the inter-domain trust curve synchronizes only with the in-domain trust value of a single system, indicating it is only related to the system conducting inter-domain accesses. As shown in (b), the fluctuation trend of inter-domain trust value lies between the two in-domain trust curves, indicating its change is correlated with both systems. Therefore, different entities within the same security domain may mutually influence each other through the global trust.

Figure 10 shows the change in DITKG in different scenarios at *t*=500s. Due to the diversity in access objects, the four graphs are totally different in both structure and transient attribute value. Taking (b) as an example, both AFC and PIS request inter-domain accesses, thus it provides the most information about the state of the whole network. Therefore, DITKG is an effective way to reflect the time-varying characteristics of global network state of intermodal transportation.

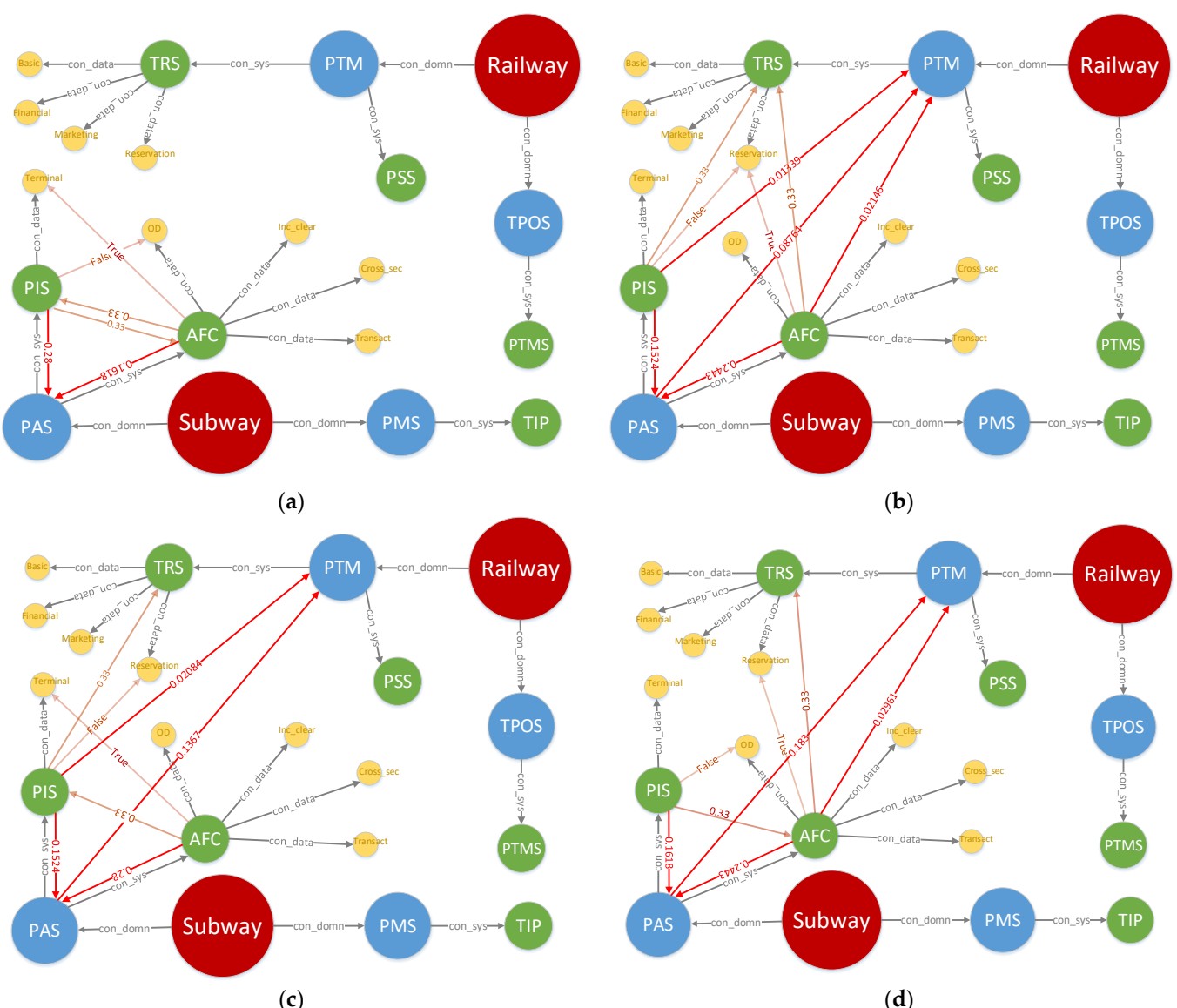

**Figure 10.** Visualization of DITKG when *t*=500s in different scenarios: (**a**) AFC and PIS both access in-domain data; (**b**) AFC and PIS both access PTM domain data; (**c**) AFC accesses in-domain data while PIS accesses PTM domain data; (**d**) PIS accesses in-domain data while AFC accesses PTM domain data.

## 7. Conclusions and Future Work

This paper focused on the comprehensive trustworthiness between different entities, aiming to break down the data silos and protect the interaction during the operation of intermodal transportation. According to the challenging demand of security and granularity, the hierarchical framework is put forward to evaluate the trust value of different types of the entities in the whole network, where data quality is regarded as a significant part of data trust. Furthermore, a DITKG based on a temporal knowledge graph is discussed as a visualization method of cross-domain interaction. By proving experimentation, the framework can satisfy the demand of data sharing in intermodal transportation. Next, we will validate the data exchange process with real data through the construction of a prototype system. Compared with related work, the results in Table 7 show that this Comprehensive trust assessment framework has advantage in data quality evaluation, hierarchical structure, dynamic updating and visualization. Therefore, the comprehensive trust assessment framework can better meet the data supervision demand of intermodal transportation.

Table 7. **The characteristics comparison** of comprehensive trust assessment mechanism in this paper with related work.

| | Data Quality | Hierarchical Structure | Dynamic Update | Visualization |
|---|---|---|---|---|
| Multilevel quality model [2] | ✓ | | | ✓ |
| Service trust model [3] | ✓ | ✓ | | |
| Information-flow-theory based model [4] | | | ✓ | |
| Block-chain-based model [5] | | | ✓ | |
| Trust-based logical model [6] | | ✓ | ✓ | |
| Comprehensive trust assessment | ✓ | ✓ | ✓ | ✓ |

As for future work, there are two aspects in sight. In the short-term, we intend to delve into network security methods and explore how to integrate appropriate secure communication strategies into our framework. This can not only enhance the security of our framework but also improve efficiency; thus, our research can better meet the needs of practical applications while ensuring data security. For our long-term plan, we will take a broader view of the intermodal transport and carry out multidisciplinary research topics, such as urban planning, environmental considerations, human behavior and larger socio-economic issues.

**Author Contributions:** Conceptualization, Z.M. and Y.W.; methodology, X.G. and P.D.; validation, X.G. and K.X., P.D. and F.K.; formal analysis, F.K.; investigation, X.G. and P.D.; resources, X.G. and F.K.; data curation, K.X. and P.D.; writing—original draft preparation, X.G.; writing—review and editing, K.X., Z.M. and Y.W.; visualization, F.K.; supervision, K.X., Z.M. and Y.W.; project administration, F.K., K.X. and P.D. All authors have read and agreed to the published version of the manuscript.

**Funding:** The work was supported by Important Project of Scientific and Technical Exploitation Program of CHINA RAILWAY (No. N2023S002).

**Data Availability Statement:** The data presented in this study are available on request from the corresponding author.

**Acknowledgments:** The authors wish to thank the reviewers for their valuable comments and suggestions concerning this manuscript.

**Conflicts of Interest:** Authors Xin Geng, Zhisong Mo, Peng Dong and Fanpeng Kong were employed by the companies China Railway Information Technology Group Co., Ltd. and China State Railway Group Co., Ltd. The remaining authors declare that the research was conducted in the absence of any commercial or financial relationships that could be construed as a potential conflict of interest.

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
