# Peer review of "On the Evaluation Framework of Comprehensive Trust for Data Interaction in Intermodal Transport"

_electronics, doi:10.3390/electronics13081487_

Round 1

Reviewer 1 Report

Comments and Suggestions for Authors

This paper proposes an interesting evaluation framework to enhance data quality and trustworthiness. The results are interesting and well-written. Some comments are as follows.

1.         What are the difficulties of the proposed evaluation framework, especially during its design.

2.         Please give more discussion on the limitations on the existing works, based on which the authors present the mechanism design of the evaluation framework.

3.         For a completed literature, some control methods also investigate secure communication framework, for example, Event-Triggered Tracking Control for Nonlinear Systems With Prescribed Performance; Event-triggered tunnel prescribed control for nonlinear systems, in which only 1-bit signal is transmitted instead of the sensitive information.

4.         Please discussion how to select the parameters of Eq 10 in the evaluation strategy.

Comments on the Quality of English Language

Some typos and grammar errors scattered in the manuscript should be corrected.

Author Response

Special thanks to you for your good comments. We tried our best to improve the manuscript and made some changes in the manuscript. These changes will not influence the content and framework of the paper. We appreciate for your warm work earnestly, and hope that the correction will meet with approval. Once again, thank you very much for your comments and suggestions.

Reviewer 2 Report

Comments and Suggestions for Authors

Concerns about research scope and significance

The research primarily emphasizes the technical elements of data reliability and quality in intermodal transportation, perhaps neglecting the larger socio-economic issues that are also important in this domain. Given the current context where transportation poses challenges not just in terms of technology but also in social and environmental aspects, it is worth noting that the article fails to adequately consider these wider consequences.

Concerns about the methods

The suggested system takes a comprehensive look at data quality and trustworthiness, but it's not clear how it could be used in real life. Some transportation groups might not be able to use multi-level assessment frameworks and trust models because they are too complicated and could use a lot of resources. This is especially true in places with poor technology infrastructure. There is insufficient discussion about how to make such a complex system work in a wide range of real-world situations without breaking the bank.

Concerns about the Data Privacy and Security Issues

The paper discusses data security and trust at length but seems to give less attention to data privacy concerns. The paper's lack of attention to privacy considerations is a notable limitation, considering the growing worldwide emphasis on user privacy and data protection requirements. The unresolved issue of balancing data sharing for operational efficiency and protecting individual privacy rights poses a significant gap in the framework's practicality in real-world situations.

Concerns about validation and empirical evidence 

Although the study incorporates practical application and simulation analysis, it is evident that there is a significant absence of empirical validation using real-world data. The lack of rigorous empirical testing might undermine the trustworthiness of the results reached, given the heavy dependence on theoretical models and simulations. Actual instances of real-world scenarios or first test runs would significantly bolster the trustworthiness and practicality of the suggested framework.

Concerns about interdisciplinary integration

The study primarily adopts a data-centric perspective, which is obviously crucial but also relatively limited in its reach. Transportation systems are fundamentally multidisciplinary, including not just data and technology, but also urban planning, environmental considerations, and human behavior. The absence of an interdisciplinary approach in both the examination and suggested remedies is a wasted chance for the comprehension and resolution of the difficulties in intermodal transportation.

Concerns about conclusion and future directions

Overall, the study provides a technically rigorous and comprehensive method for assessing data quality and trust in intermodal transportation systems. However, it does not adequately include wider practical, socio-economic, and privacy issues. Future research in this field should adopt a comprehensive strategy that includes empirical validations, takes into account practical implementation issues, and incorporates multidisciplinary viewpoints to effectively tackle the intricacies of contemporary transportation systems.

Author Response

(The authors gave the same response as above.)

Round 2

Reviewer 1 Report

Comments and Suggestions for Authors

All my concerns have been addressed.

Comments on the Quality of English Language

Minor editing of English language required